# Public perceptions of synthetic cooling agents in electronic cigarettes on Twitter

**Andrew H. Liu**[1], **Julia Hootman**[2], **Dongmei Li**[3‡]*, **Zidian Xie**[3‡]

**1** Department of Computer Science, University of North Carolina at Chapel Hill, Chapel Hill, North Carolina, United States of America, **2** Department of Computer Science, Xavier University, Cincinnati, Ohio, United States of America, **3** Department of Clinical and Translational Research, University of Rochester Medical Center, Rochester, New York, United States of America

☯ These authors contributed equally to this work.
‡ DL and ZX also contributed equally to this work.
* Dongmei_Li@urmc.rochester.edu

**Data Availability Statement:** The Python code used for Twitter/X data analysis is publicly available from the Github website https://github.com/ahl1u/PLOSOne-Ecig. The Twitter data are publicly

## Abstract

Amid a potential menthol ban, electronic cigarette (e-cigarette) companies are incorporating synthetic cooling agents like WS-3 and WS-23 to replicate menthol/mint sensations. This study examines public views on synthetic cooling agents in e-cigarettes via Twitter data. From May 2021 to March 2023, we used Twitter Streaming Application Programming Interface (API), to collect tweets related to synthetic cooling agents with keywords such as 'WS-23,' 'ice,' and 'frozen.' The deep learning RoBERTa (Robustly Optimized BERT-Pre-training Approach) model that can be optimized for contextual language understanding was used to classify attitudes expressed in tweets about synthetic cooling agents and identify e-cigarette users. The BERTopic (a topic modeling technique that leverages Bidirectional Encoder Representations from Transformers) deep-learning model, specializing in extracting and clustering topics from large texts, identified major topics of positive and negative tweets. Two proportion Z-tests were used to compare the proportions of positive and negative attitudes between e-cigarette users (vapers) and non-e-cigarette-users (non-vapers). Of 6,940,065 e-cigarettes-related tweets, 5,788 were non-commercial tweets related to synthetic cooling agents. The longitudinal trend analysis showed a clear upward trend in discussions. Vapers posted most of the tweets (73.05%, 4,228/5,788). Nearly half (47.87%, 2,771/5,788) held a positive attitude toward synthetic cooling agents, which is significantly higher than those with a negative attitude (19.92%,1,153/5,788) with a P-value < 0.0001. The likelihood of vapers expressing positive attitudes (60.17%, 2,544/ 4,228) was significantly higher ($P < 0.0001$) than that of non-vapers (14.55%, 227/1,560). Conversely, negative attitudes from non-vapers (30%, 468/1,560) were significantly ($P < 0.0001$) higher than vapers (16.2%, 685/4,228). Prevalent topics from positive tweets included "enjoyment of specific vape flavors," "preference for lush ice vapes," and "liking of minty/icy feelings." Major topics from negative tweets included "disliking certain vape flavors" and "dislike of others vaping around them." On Twitter, vapers are more likely to have a positive attitude toward synthetic cooling agents than non-vapers. Our study provides important insights into how the public perceives synthetic cooling agents in e-

available from the Twitter/X website: https://twitter.com/home?lang=en.

**Funding:** This study was supported by REU Site: Computational Methods for Understanding Music, Media, and Minds grant funded by National Science Foundation (Award number: 1950460) (AL and JH) and by the WNY Center for Research on Flavored Tobacco Products (CRoFT) under cooperative agreement U54CA228110 funded by National Cancer Institute and US Food and Drug Administration (FDA) (DL and ZX). The content is solely the responsibility of the authors and does not necessarily represent the official views of the NIH or the FDA. The funders had no role in study design, data collection and analysis, decision to publish, or preparation of the manuscript.

**Competing interests:** All authors have no potential conflict of interest to declare.

cigarettes. These insights are crucial for shaping future U.S. Food and Drug Administration (FDA) regulations aimed at safeguarding public health.

## Introduction

Electronic cigarettes (e-cigarettes) are devices used to aerosolize a solution typically containing propylene glycol, vegetable glycerin, flavorings, and, in most cases, nicotine for inhalation by consumers [1, 2]. The prevalence of e-cigarette use has risen exponentially in the last decade, especially in youth and young adults [3, 4]. E-cigarettes have become the most frequently used tobacco product among high school students, with an estimated 3.6 million middle and high school students reporting current e-cigarette use in 2022 [5]. Toxicity and health impact studies of e-cigarettes have shown that components in e-cigarette aerosols could lead to severe adverse health problems, including respiratory disorders, cardiovascular diseases, mental health issues, and possibly cancer [6–9]. In February 2020, the U.S. Food and Drug Administration (FDA) issued the flavor enforcement policy to limit the sale of cartridge-based e-cigarette products to only tobacco and menthol-flavored e-cigarettes [10]. Following the announcement and implementation of the FDA's new policy, there was a significant increase in the sales of menthol-flavored e-cigarettes [11]. To curb the increasing popularity of menthol-flavored e-cigarettes, the New York State Department of Health banned all flavored vaping products, including menthol-flavored e-cigarettes, starting from May 18, 2020 [12]. Menthol-flavored cigarettes have been popular, with 40% of smokers using menthol-flavored cigarettes in 2020 [13]. In April 2022, the FDA announced the proposed standard to ban menthol–the last allowable flavor–in cigarettes in hopes of reducing tobacco-related diseases and deaths [14].

In response to the proposed menthol cigarette ban and potential national menthol-flavored e-cigarette ban, the market has seen a significant increase in 'ice' flavors that attempt to circumvent these regulations by incorporating synthetic cooling agents like WS-3 and WS-23 into e-cigarette refill liquids to mimic the cooling sensation like menthol or mint [15]. Ice-flavored e-cigarette use correlates with higher vaping frequency and dependence, and 44.8% of e-cigarette users primarily use ice flavors [16]. Synthetic cooling agents in e-cigarettes imitate menthol's cooling sensation but induce a unique 'cold' feeling, akin to the feeling of inhaling frigid air, during inhalation, unlike the 'hot' burning or spicy sensation from capsaicin or menthol's 'cool', mild and refreshing mint feeling [17]. These agents are now prevalent in 'ice' flavored e-cigarettes, despite many consumers' intake of the agents exceeding established safety thresholds [18, 19]. This increasing popularity of synthetic cooling agents creates new concerns about the potential health risks posed by industry attempts to bypass regulations [18]. While the long-term health effects of synthetic cooling agents are largely unknown due to their relative newness, one recent study showed that synthetic cooling agents may disrupt respiratory airway epithelial cell (AEC) responses, a vital defense against inhaled toxins and allergens [20]. This disruption could impact the physiology of the airway and impact susceptibility to health-threatening respiratory diseases [20].

Understanding the public perceptions and discussions of synthetic cooling agents is crucial for shaping future regulation. Social media platforms, notably Twitter (now rebranded as "X"), have been widely used in capturing public perceptions and discussions of tobacco products [21]. As of mid-2023, Twitter has over 235 million daily active users, with an average of 500 million tweets posted daily [22]. This massive volume of interactions has made it an important resource for investigating public perceptions and discussions on various topics, including

health and lifestyle behaviors like vaping. For instance, researchers analyzed thousands of tweets to investigate attitudes toward e-cigarette flavors and found that preferences for certain flavors, such as fruit/fruit beverage, dessert/pastry, and candy, chocolate, or sweets, might be associated with continued e-cigarette use [23]. Twitter has also been employed to study the perceived health impacts of vaping [24]. A study analyzing Twitter posts found that many e-cigarette users reported negative health effects, ranging from minor complaints to more serious conditions [25]. A cross-sectional study on marketing of electronic cigarettes on Twitter highlighted Twitter's value in tracking real-time public sentiment [26]. Overall, Twitter has served as an ideal resource for understanding public perceptions and discussions of synthetic cooling agents.

Our study aims to examine the public perceptions and discussions of 'ice' flavored e-cigarettes and associated synthetic cooling agents using Twitter data. Using natural language processing and deep-learning algorithms, our study results will provide a snapshot of public discourse and attitudes towards 'ice' flavored e-cigarettes and synthetic cooling agents. Our findings will provide vital insights into the dynamic e-cigarette landscape, specifically in relation to 'ice' flavors and synthetic cooling agents. Understanding these dynamics can inform future regulations and strategies to protect public health. This aim was successfully achieved, as evidenced by our detailed analysis and significant results, which provide a comprehensive understanding of public sentiment regarding synthetic cooling agents in e-cigarettes.

## Materials and methods

### Ethics statement

This study was reviewed and approved by the Research Subjects Review Board of the Office for Human Subject Protection at the University of Rochester (STUDY00006570). Patient consent for publication is not required as the data were analyzed anonymously.

### Data collection and preprocessing

For this study, we leveraged Twitter data from the Twitter Streaming API (Application Programming Interface) collected from May 3, 2021, to March 14, 2023 using keywords related to electronic cigarettes [27]. Tweets were initially filtered based on keywords related to e-cigarettes, including but not limited to keywords like "e-cig", "vaping", "e-liquid", and "vapenation". After filtering out the retweets from e-cigarette related tweets, we obtained 2,597,206 tweets. We applied another layer of filter to filter out promotional content with promotional keywords within the username, retweets, and content of the tweet from further analysis [27]. This resulted in a collection of 2,426,615 non-promotional tweets. We applied a third layer of filter to filter out "ice" flavored e-cigarettes or synthetic cooling agents related tweets using keywords like 'ice', 'frozen', and 'arctic' (S1 Table) and ultimately identified 5,788 tweets related to synthetic cooling agents and sensations (S1 Fig).

### Deep learning models for classification

We used an inductive method to manually label 2,000 randomly selected tweets, which served as the training data for our deep learning classification models. We first randomly selected 200 tweets and manually labeled them according to the users' attitudes toward synthetic cooling agents and sensations into negative, neutral, and positive categories. Additionally, we determined whether the Twitter users were e-cigarette users (vapers), indicated from the tweets. Two authors (AL and JH) hand-coded the 200 tweets independently and reached a high inter-rater reliability (Cohen's Kappa coefficient was 0.9219 for attitude and 0.9256 for whether the

Twitter user was an e-cigarette user). Then, the remaining 1,800 tweets were manually coded by the same two authors (AL and JH) with each coding 900 tweets.

We built two RoBERTa (Robustly Optimized BERT Pre-Training Approach) models using the 2,000 manually coded tweets as the training dataset [28], one for sentiment analysis and the other for binary classification to determine whether a Twitter user was an e-cigarette user. RoBERTa is a deep learning model developed by Facebook AI, which builds upon BERT (Bidirectional Encoder Representations from Transformers) [29]. This model differs from its predecessor by enhancing the pretraining process, involving increased training data and a more dynamic masking pattern, leading to improved accuracy on various natural language processing tasks. We employed ten-fold cross-validation to ensure the robustness of our models. Our models achieved an F1-score of 0.83 for attitude and 0.92 for e-cigarette user classification, ensuring the validity of our model results. The two-proportion Z-test was used to compare differences in the proportions of various sentiment categories.

### Topic modeling

To understand the topics discussed in tweets related to synthetic cooling agents and sensations, we employed BERTopic [30]. BERTopic is an advanced, context-aware topic modeling technique. BERTopic is based on the BERT language model, which allows us to grasp the larger context of the conversation by capturing semantic relationships between words and topics. We determined the optimal number of topics based on the coherence score and inter-topic distances [31, 32].

## Results and discussion

### Public discussions on synthetic cooling agents, sensations, and e-cigarettes on Twitter

We observed a clear increasing trend in the number of tweets related to ice flavors or synthetic cooling agents in e-cigarettes from May 2021 to March 2023 with some variations (Fig 1). In addition, there is an increasing trend in the proportion of tweets related to synthetic cooling agents among all e-cigarette-related tweets over time (S2 Fig).

Among 5,788 tweets related to synthetic cooling agents and sensations in e-cigarettes, nearly half had a positive sentiment (47.81%), about a fifth were negative (19.92%), and the remaining one third (32.20%) had a neutral sentiment. The proportion of tweets with a positive attitude was significantly higher than that with a negative attitude (P-value <0.0001).

As shown in Fig 2, the proportion of tweets with either a positive or negative attitude from May 2021 to April 2023 was relatively consistent. While it is not significant, there is a small increase in the proportion of negative tweets and a slight decrease in the proportion of positive tweets over time. Among 3,924 tweets related to synthetic cooling agents and sensations in e-cigarettes with positive or negative sentiment (omitting neutral), the majority had a positive sentiment (70.62%) with a smaller proportion being negative (29.38%). The proportion of tweets with a positive attitude is significantly higher than that with a negative attitude when observing only tweets of positive and negative sentiment (P-value <0.0001).

### Twitter users' attitudes toward synthetic cooling agents and sensations between vapers and non-vapers

From 5,788 tweets related to synthetic cooling agents and sensations, we have identified 5,067 unique Twitter users. Among them, most (74.15%) are e-cigarette users (vapers) with the remaining (25.85%) being non-e-cigarette users (non-vapers). As shown in S3 Fig, the

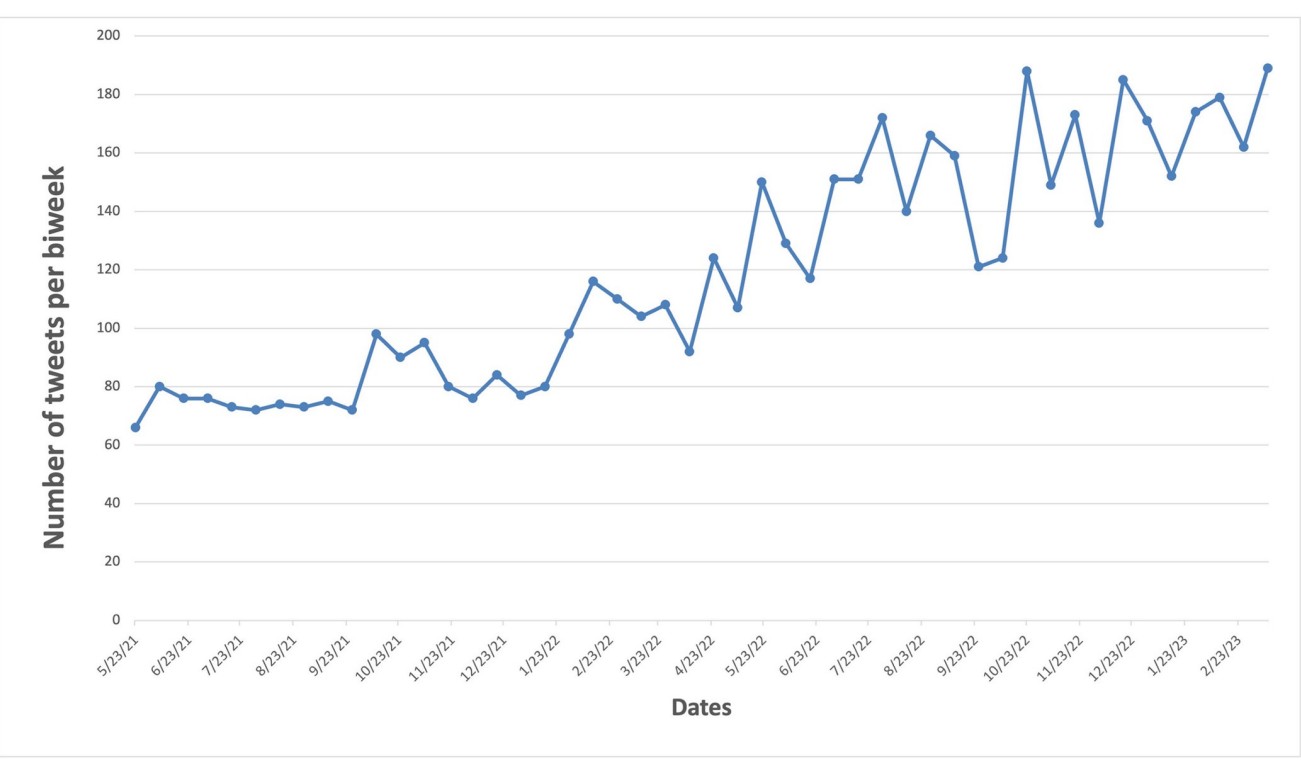

**Fig 1. The number of tweets on synthetic cooling agents and sensations over time.**

longitudinal proportion of e-cigarette users and non-e-cigarette users from May 2021 to March 2023 has been relatively consistent. Vapers (n = 3,757) predominantly expressed positive attitudes in their tweets (59.70%). In addition, a smaller proportion (23.13%) of vapers expressed neutral attitudes, and an even smaller proportion (17.17%) expressed negative attitudes. Nearly half of all non-vapers (n = 1,310) expressed neutral attitudes (49.77%). More than a third non-vapers showed negative attitudes (34.73%) and less than one sixth non-vapers showed positive attitudes (15.50%).

The comparison of frequency distributions revealed that Twitter users' attitudes toward synthetic cooling agents and sensations varied between vapers and non-vapers (Fig 3). Vapers displayed more positive attitudes (59.70%) than non-vapers (15.50%). Conversely, non-vapers exhibited more negative attitudes (34.73%) than vapers (17.17%). Additionally, non-vapers (49.77%) showed more neutral attitudes than vapers (23.13%).

## Public discussions of synthetic cooling agents and sensations in e-cigarettes on Twitter

Table 1 showed the major topics discussed about the synthetic cooling agents and sensations in e-cigarettes. Among tweets with positive attitude (n = 2771), the most prevalent topic is the enjoyment of ice vapes (64.49%), followed by liking lush ice vapes (18.00%), liking the minty or icy feeling (7.32%), certain flavor being refreshing (3.10%), and enjoying the smell of vapes (0.79%).

Table 1 also highlights the main issues brought up in tweets with a negative attitude (n = 1,153) towards the synthetic cooling agents and sensations in e-cigarettes. The most

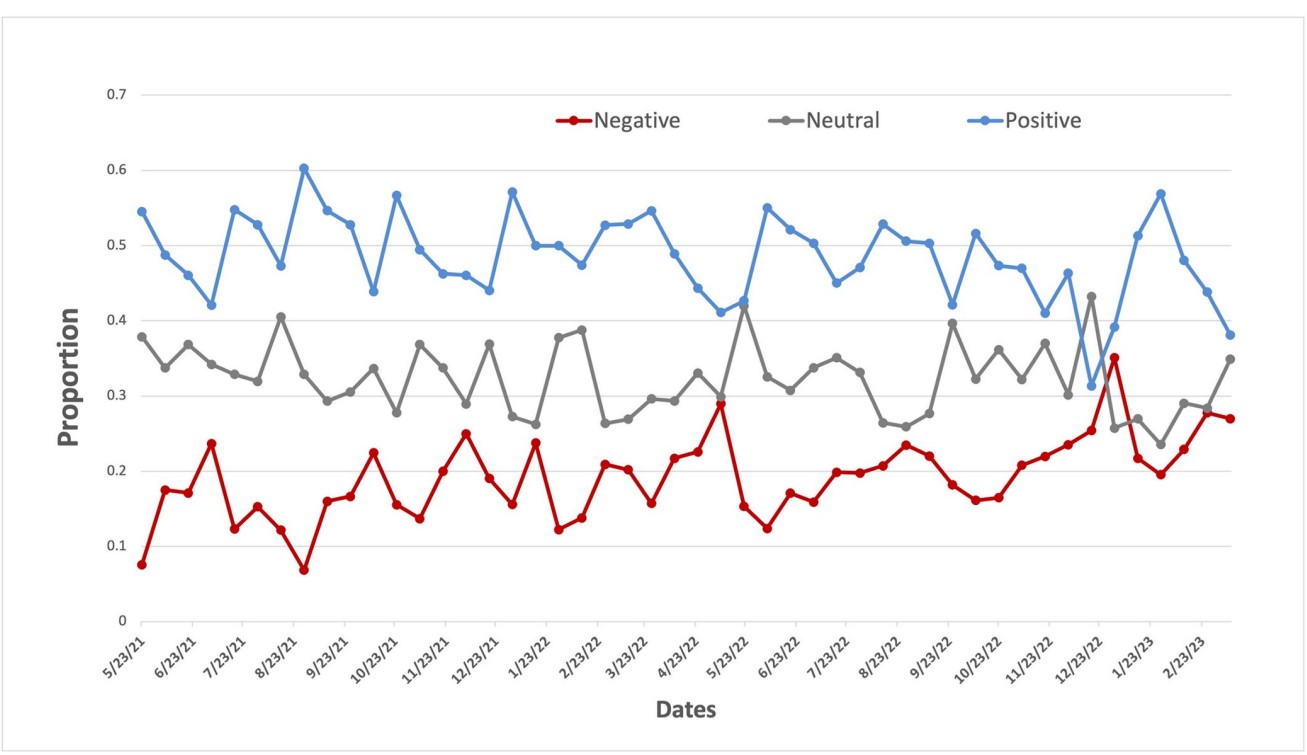

**Fig 2. The attitude toward synthetic cooling sensation on Twitter over time.**

recurrent topic is disliking certain vape flavors (53.69%), followed by discomfort from others vaping around them (20.82%), disagreement with the act of vaping itself (6.42%), and disliking the mintiness of vapes (4.51%).

We identified an increasing trend in the discussions of the synthetic cooling agents and sensations in e-cigarettes from May 2021 to March 2023. Interestingly, the public attitudes toward the synthetic cooling agents and sensations in e-cigarettes remains relatively constant over the study period. Furthermore, we found a predominantly positive attitude toward synthetic cooling agents and sensations in e-cigarettes. Further analysis of Twitter users revealed that this positive sentiment was largely driven by e-cigarette users, or vapers. The topics discussed within these positive tweets mainly centered on the enjoyment of ice-flavored vaping. Though, the importance of flavor variety in vaping experiences was also noted. Conversely, the primary concerns arising from tweets with negative attitudes involved second-hand exposure to vaping and potential addiction.

Consistent with our findings of a mostly positive attitude towards synthetic cooling agents and sensations in e-cigarettes, a previous content analysis of ice-flavored e-cigarette products using Twitter data from January to July 2021 also revealed a majority favorable attitude towards these products [33].

We observed an increasing trend in the discussions of the synthetic cooling agents and sensations during the study period, which might partially be due to the aggressive marketing of ice-flavored e-cigarette products identified from the previous study [33]. Another reason for this increasing trend might be due to their increased popularity and conversations related to their regulations. For example, in May 2022 we saw an increase in tweets responding to a television anti-vaping commercial with commentary like "Did they just use Frost clips for a Stop

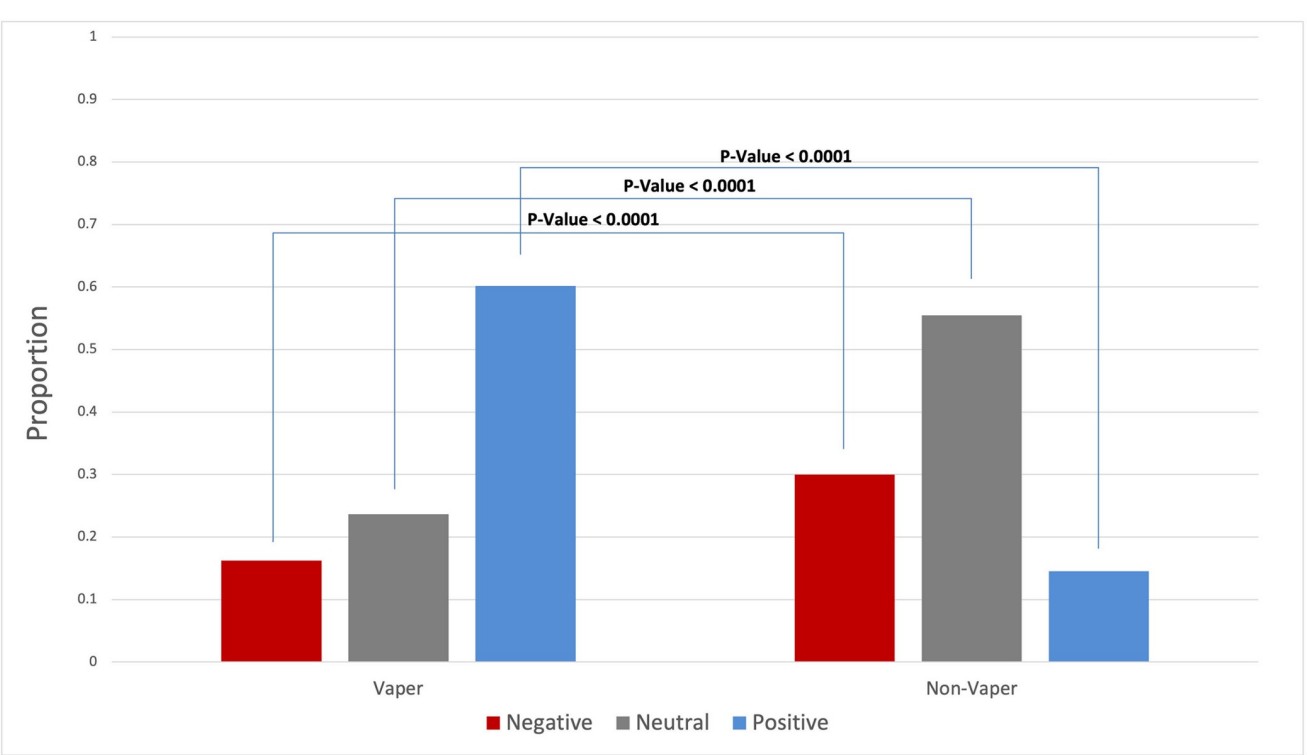

**Fig 3. Comparison of attitudes toward synthetic cooling agents and sensations between vapers and non-vapers.**

Vaping commercial?" and "Killer Frost cape commercial makes me want to vape!" These responses correspond with an increase in discussions of synthetic cooling agents in May 2022. The increasing trend suggests that synthetic cooling agents are becoming more prevalent and widely used, which may be due to their perceived benefits or effectiveness in providing a pleasant experience [34]. However, there is also growing concern about their potential health risks, leading to increased discussions about policy and regulation. Studies have already shown that the amount of synthetic cooling agent, WS-3/WS-23, in e-cigarettes often cause levels of exposure higher than the safety threshold [18, 34]. Elevated exposure to WS-3/WS-23 has been linked to changes in airway epithelial cells, potentially increasing susceptibility to respiratory diseases [20]. Although more research is needed on the broader health effects that these synthetic agents may have, preliminary findings suggest a need for immediate regulatory attention.

The comparison of attitudes between vapers and non-vapers towards synthetic cooling agents and sensations indicated that a higher percentage of vapers expressed positive sentiments, while non-vapers displayed more negative and neutral sentiments. This underlines the divergent views between these two groups and can be attributed to their direct experiences and exposure to synthetic cooling agents, especially enjoyment of specific vape flavors and preference for certain vape types. For example, tweets like "I don't vape until someone pulls out lush ice" emphasize the allure of specific mixtures like Lush Ice, which offers a mix of watermelon and melon flavors with a minty kick. This contrasts with standard ice vapes, known for a straightforward cooling sensation. Tweets like "Told mum I like vaping for the minty flavor and she goes I'll make you a mint chutney pls stop," underline the minty flavors' appeal. Additionally, mentions of enjoying the vapor smell, such as in the tweet "hi moots I ran out of my

**Table 1. Major topics identified from tweets with positive or negative attitudes.**

| Attitude | Topic | Percentage of Tokens, n (%) | | Examples |
|---|---|---|---|---|
| Positive (n = 2,771) | Enjoyment of ice vapes | 1,787 (64.49%) | 'ice', 'vape', 'banana', 'peach', 'vapes', 'strawberry', 'watermelon', 'like', 'flavor', 'good' | 'Me yesterday *damn I need to quit vaping again* My brain today *peach ice vape go brrrr*' 'I need banana ice vape juice in a 50MG please' |
| | Liking lush ice vapes | 499 (18.00%) | 'ice', 'vape', 'vaping', 'iced', 'vapes', 'like', 'im', 'juice', 'menthol', 'lush' | 'This lush ice vape is my new fave' 'don't vape until someone pulls out lush ice' |
| | Liking the minty/icy feelings | 203 (7.32%) | 'minty', 'mint', 'vape', 'like', 'feel', 'fresh', 'vapes', 'taste', 'hitting', 'flavor' | 'Just vape straight menthol like me I can't even get Cancer anymore my lungs are so mint' 'Sounds lovely. My Mrs would like that. She loves her minty Vapes mate' |
| | Certain flavors being refreshing | 86 (3.10%) | 'refreshing', 'cooling', 'vaping', 'vape', 'refreshed', 'cigarette', 'day', 'feel', 'coolness', 'like' | 'I rather like my Heisenberg Menthol vape. It's as refreshing as a brisky mountain's flowing river.' 'Blue raspberry chilled is such a nice refreshing fruit flavour' |
| | Enjoying the smell of vapes | 22 (0.79%) | 'smell', 'like', 'room', 'strawberry', 'bathroom', 'ice', 'public', 'black', 'rather', 'vape' | 'my mom come in the bathroom after me bd be like wow this plugin smells so nice… baby that was my strawberry ice vape' 'life hack: if u vape inside ur closet all ur clothes end up smelling like ice berry bomb [smiling face with hearts x3]*' |
| Negative (n = 1,153) | Disliking certain vape flavors | 619 (53.69%) | 'ice', 'vape', 'like', 'banana', 'taste', 'flavor', 'vapes', 'shit', 'peach', 'smell' | 'my banana ice vape tastes like someone put mint toothpaste on a banana laffy taffy [vomit]*' 'my biggest regret is convincing myself that mango ice vapes taste good' |
| | Disliking others vaping around them | 240 (20.82%) | 'vape', 'ice', 'like', 'vaping', 'im', 'vapes', 'face', 'strawberry', 'mf', 'iced' | 'I hate people that vape, like I could be talking to someone and all of a sudden lush ice is being blown into my face' 'manwho bought ice was vaping in front me… no sir pls get that sh*t away from me [unamused face]*' |
| | Disagreement with the act of vaping | 74 (6.42%) | 'addicted', 'embarrassing', 'blast', 'grip', 'cherry', 'mango', 'f***', 'get', 'vaping' | 'You're hooked on that Tango Cherry Ice Blast flavour. Get help—vaping is unprofessional.' 'Being addicted to vaping must be embarassing, how do you get addicted to strawberry icy cold fresh vape' |
| | Disliking vapes due to mintiness | 52 (4.51%) | 'minty', 'mint', 'vapes', 'vape', 'like', 'taste', 'flavored', 'icy', 'feel', 'im' | 'Just sucked on Jamie's vape (not a euphemism) and nearly died a minty fresh death' 'My mom left her vape in the bathroom so I tried using it and bro it feels minty why do ppl vape?' |

*Indicates the presence of an emoji

disposable vape and now I miss my ice grape-smelling room," show that sensory enjoyment isn't limited to taste.

The recurring theme of addictiveness and pleasurable sensations highlights the sensory and innovative appeal of vaping. This trend, reflecting a growing acceptance or normalization of vaping within certain communities, has crucial implications for future policy. Policymakers may need to address these specific attractions when instituting regulations, ensuring that marketing and promotion of vaping products do not exploit these preferences.

Tweets with negative attitudes expressed dislike of certain flavors and concerns about second-hand exposure. Examples like "@mentholdiet banana ice OR literally anything but watermelon (watermelon vapes make me vomit in a bad way: /)" and "Minty vape flavors give me major brain freeze vibes. I can't do it," point to specific aversions to flavors, ranging from mint to fruit. These dislikes could be due to individual taste preferences or sensations like "brain freeze," contrasting with a broader trend of users finding non-traditional flavors like fruit or candy more satisfying [35]. Concerns about second-hand exposure also resonate strongly. Tweets such as "you blow peach ice in my face and watch how i shove that vape down your throat," or "Vape smokers will blow banana ice in ya face while u telling em how ya grandma died," spotlight the intrusion of vapor into personal spaces, exacerbating existing concerns about the inhalation of harmful substances such as nicotine and toxic chemicals by bystanders [36–38].

These findings provide insight into the specific factors that influence people's perceptions of synthetic cooling agents, highlighting concerns about health, etiquette, and the social impact of vaping. As evidence, social media posts like these offer a rich source of qualitative data that traditional surveys may not capture. The immediate and personal nature of these posts provides valuable context. While there are many predictable tweets about frequently used products like popular vape flavors, Twitter also captures anecdotal views, which can provide a broader perspective. Unlike surveys, which typically present structured questions, social media data reveals spontaneous, unfiltered opinions and real-time discussions [38].

Our study has several limitations. Despite demonstrating the impressive performance of the deep-learning models we employed for tweet labeling, they are not flawless and could potentially introduce some misclassification and bias into our findings. Furthermore, the absence of demographic data (such as age, gender, and race/ethnicity) for the Twitter users in our dataset restricts our ability to explore the attitudes and topics discussed based on each demographic subgroup. We also deliberately avoided using the term "cool" as a keyword to minimize the collection of irrelevant tweets. While this decision reduced noise, it also likely omitted some relevant tweets discussing synthetic cooling agents in a relevant context. Moreover, people being more inclined to share their strong opinions on social media can lead to potential overrepresentation of users that have extremely negative or extremely positive sentiments. We also did not account for the potential influence of bot accounts, which could have skewed both the sentiment and the topical focus of the tweets collected. Furthermore, the longitudinal nature of our data collection doesn't consider temporal shifts in public opinion that could be influenced by new research findings or policy changes. In addition, although we filtered out commercial content using related keywords in the preprocessing step, a small proportion of advertisement-related tweets, primarily sponsored YouTube review tweets and seller promotions that didn't contain exclusion keywords, were included in the dataset. However, their presence was negligible and did not significantly impact the results of our analysis. Lastly, our study is constrained by its focus on English-language tweets and its reliance on Twitter as the sole platform, which may not fully represent global and/or multilingual perspectives.

## Conclusions

Using Twitter data, we investigated public perceptions and discussions of synthetic cooling agents and sensations in e-cigarettes. Our analysis revealed an increasing trend in discussions about these agents and sensations over time, with positive attitudes consistently dominating the study period. Vapers were found to be the major contributors to the overall positive attitudes, expressing their preference for ice-flavored e-cigarette products. Tweets with negative attitudes expressed dislike for certain flavors and concerns about second-hand vapor exposure. Our results provide valuable information for future regulation of synthetic cooling agents. The favorable view of ice-flavored e-cigarette products raises significant public health concerns, given the potential harms of these synthetic cooling agents. Public health campaigns are needed to educate the public about the potential risks associated with synthetic cooling agents. These campaigns could be delivered through popular social media platforms such as Twitter to reach a large and engaged audience.

## Supporting information

**S1 Fig. Flowchart of Twitter data pre-processing.**
(TIF)

**S2 Fig. The proportion of tweets related to synthetic cooling agents among all e-cigarette-related tweets on Twitter.**
(TIF)

**S3 Fig. The proportion of vapers and non-vapers mentioning synthetic cooling agents and sensations on Twitter.**
(TIF)

**S1 Table. Keywords used in Twitter data filtering.**
(DOCX)

## Author Contributions

**Conceptualization:** Dongmei Li, Zidian Xie.

**Data curation:** Zidian Xie.

**Formal analysis:** Andrew H. Liu, Julia Hootman.

**Funding acquisition:** Dongmei Li.

**Investigation:** Zidian Xie.

**Methodology:** Andrew H. Liu, Julia Hootman, Zidian Xie.

**Project administration:** Zidian Xie.

**Resources:** Zidian Xie.

**Supervision:** Dongmei Li, Zidian Xie.

**Validation:** Andrew H. Liu.

**Visualization:** Andrew H. Liu, Julia Hootman.

**Writing – original draft:** Andrew H. Liu, Julia Hootman, Dongmei Li, Zidian Xie.

**Writing – review & editing:** Andrew H. Liu, Julia Hootman, Dongmei Li, Zidian Xie.

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
