## [Decision Letter · Decision Letter 0]

17 Jan 2024

PONE-D-23-30383Public perceptions of synthetic cooling agents in electronic cigarettes on TwitterPLOS ONE

Dear Dr. Li,

Thank you for submitting your manuscript to PLOS ONE. After careful consideration, we feel that it has merit but does not fully meet PLOS ONE’s publication criteria as it currently stands. Therefore, we invite you to submit a revised version of the manuscript that addresses the points raised during the review process.

We look forward to receiving your revised manuscript.

Kind regards,

Gea Oliveri Conti, Ph.D. MBs

Academic Editor

PLOS ONE

“This study was supported by REU Site: Computational Methods for Understanding Music, Media, and Minds grant funded by National Science Foundation (Award number: 1950460) (AL and JH) and by the WNY Center for Research on Flavored Tobacco Products (CRoFT) under cooperative agreement U54CA228110 funded by National Cancer Institute and US Food and Drug Administration (FDA) (DL and ZX). The content is solely the responsibility of the authors and does not necessarily represent the official views of the NIH or the FDA.”

“This study was supported by REU Site: Computational Methods for Understanding Music, Media, and Minds grant funded by National Science Foundation (Award number: 1950460) and by the WNY Center for Research on Flavored Tobacco Products (CRoFT) under cooperative agreement U54CA228110 funded by National Cancer Institute and US Food and Drug Administration (FDA). The content is solely the responsibility of the authors and does not necessarily represent the official views of the NIH or the FDA.”

“This study was supported by REU Site: Computational Methods for Understanding Music, Media, and Minds grant funded by National Science Foundation (Award number: 1950460) (AL and JH) and by the WNY Center for Research on Flavored Tobacco Products (CRoFT) under cooperative agreement U54CA228110 funded by National Cancer Institute and US Food and Drug Administration (FDA) (DL and ZX). The content is solely the responsibility of the authors and does not necessarily represent the official views of the NIH or the FDA.”

6. In the online submission form you indicate that your data is not available for proprietary reasons and have provided a contact point for accessing this data. Please note that your current contact point is a co-author on this manuscript. According to our Data Policy, the contact point must not be an author on the manuscript and must be an institutional contact, ideally not an individual. Please revise your data statement to a non-author institutional point of contact, such as a data access or ethics committee, and send this to us via return email. Please also include contact information for the third party organization, and please include the full citation of where the data can be found.

Reviewers' comments:

Reviewer's Responses to Questions

**Comments to the Author**

1. Is the manuscript technically sound, and do the data support the conclusions?

Reviewer #1: Yes

Reviewer #2: Yes

2. Has the statistical analysis been performed appropriately and rigorously? 

Reviewer #1: Yes

Reviewer #2: Yes

3. Have the authors made all data underlying the findings in their manuscript fully available?

Reviewer #1: Yes

Reviewer #2: Yes

4. Is the manuscript presented in an intelligible fashion and written in standard English?

Reviewer #1: Yes

Reviewer #2: Yes

5. Review Comments to the Author

Reviewer #1: Thank you for the opportunity to review this manuscript. Overall, it provides information on an interesting approach to examining public views on synthetic cooling agents in e-cigarettes using data from Twitter (May 2021 to March 2023). This work is timely and appropriate for the journal.

Below are my comments/suggestions for improvement:

ABSTRACT

1. Minor point: There is inconsistent capitalization of vapers / non-vapers throughout. Please correct.

2. Minor point: Please spell out formal names of acronyms when they are introduced (e.g., FDA).

3. If possible, differentiate (for the reader) roBERTa vs. BERTopic models.

INTRODUCTION:

1. Minor point: Again, please spell out formal names of acronyms when they are introduced (e.g., FDA).

2. Please make it explicit to the readership that the policy limiting the sale of cartridge-based e-cigarette products to only tobacco and menthol flavored e-cigarettes is the "FDA's flavor enforcement policy".

3. It is noted that "As menthol-flavored e-cigarettes gain popularity, menthol-flavored cigarettes are also becoming increasingly popular among youth and young adults, particularly in the African American community." More context needs to be provided for this statement as it does not fit with the rest of the paragraph and/or more information is needed to explain that this issue is a contributor to tobacco-related disparities (if that is the intent). Also, menthol-flavored cigarettes have been popular before menthol flavored e-cigarettes... so more needs to be clarified in these sentences.

4. Minor point: On page 5, line 98-99, missing "has" in "Overall, Twitter (has) served as an ideal..."

METHODS:

1. The detail provided seems to be adequate.

RESULTS:

1. Figures seem to blurry - authors may need to save the files in a different format and re-upload.

2. Rather than showing the percentage and the calculation for percentages, it might help to simplify the language and indicate that: "Among tweets with a positive attitude (n=2771), the most prevalent topic is enjoyment of ice vapes (64.49%), followed by liking lush ice vapes (18.0%) - for example. Also, the reporting of decimal places does not seem to be consistent.

DISCUSSION:

1. It might be helpful if the authors can comment on whether any of the posts were #ads, as it would give more nuance regarding whether messaging is coming from vapers directly, or via the tobacco industry (for example).

2. It doesn't look like the discussion highlights a lot of other literature - is this a function of the approach being novel? If not, more citations are needed.

3. Limitations as written is appropriate.

Reviewer #2: The authors use machine learning to classify tweets with content about ice-type vape flavors. Classifications included valence, and whether the tweet was from a vaper or not.

Perhaps the most interesting finding was that the frequency of tweets that satisfied their search criteria more than doubled over the 20ish months of data. However, interpretation of this would be made easier if the reader was given comparison trajectories. Was the total tweets per month constant? What about the number of tweets about vaping in general – is this just reflecting overall change in interest, or, as plausibly suggested, a specific growth in interest in the WS-3/-23 agents.

In addition to total interest as conveyed in tweet frequency, there are things I can imagine getting out of tweet content that would not be easily captured by questionnaire data. But the authors primarily focus on whether people do or do not like ice flavors. They conclude that they mostly do. But this seems a very indirect way of assessing how people feel about these products – it requires assumptions about what the relationship is to tweet content and general opinions. For example, I suppose people tweet more about things that they have strong opinions about. I have no idea whether they are more likely to tweet about things they like a lot or things they dislike. But vapes they like a lot are going to be more common in their day, so that might lead to over-representation of positive content. Or maybe people like to tweet views they think are less common. So my question is, what is it that these data are conveying that is not better conveyed by survey data and market data? There might be convincing answers to this that could be included.

The authors write, “Synthetic cooling agents in e-cigarettes imitate menthol's cooling sensation but induce a unique 'cold' feeling during inhalation, unlike the 'hot' sensation from capsaicin or menthol's 'cool' feeling.” I read the prior tesxt to imply the WS-3 / -23 were intended to be similar to menthol to get around a potential ban. In what way is it “unlike” menthol’s cool feeling?

6. PLOS authors have the option to publish the peer review history of their article (what does this mean?). If published, this will include your full peer review and any attached files.

Reviewer #1: No

Reviewer #2: No

---

## [Author Response · Author response to Decision Letter 0]

25 Jan 2024

Overview of responses to reviewers

Manuscript ID: PONE-D-23-30383

PLOS ONE

We appreciate the many suggestions and comments from the reviewers. We have revised our manuscript to incorporate the reviewers’ comments and suggestions. The details are listed below.

Reviewer #1: Thank you for the opportunity to review this manuscript. Overall, it provides information on an interesting approach to examining public views on synthetic cooling agents in e-cigarettes using data from Twitter (May 2021 to March 2023). This work is timely and appropriate for the journal.

Below are my comments/suggestions for improvement:

ABSTRACT

1. Minor point: There is inconsistent capitalization of vapers / non-vapers throughout. Please correct. 

Response: We have corrected inconsistent capitalization of vapers/non-vapers in our revised manuscript.

2. Minor point: Please spell out formal names of acronyms when they are introduced (e.g., FDA).

Response: We have spelled out full names of the acronyms “API”, “FDA”, “RoBERTa”, and “BERTopic” for their first appearance within the Abstract section of our revised manuscript.

3. If possible, differentiate (for the reader) roBERTa vs. BERTopic models.

Response: We have added brief explanations to emphasize a distinction between the RoBERTa and BERTopic models.

INTRODUCTION:

1. Minor point: Again, please spell out formal names of acronyms when they are introduced (e.g., FDA).

Response: We have spelled out the formal name of the acronym “FDA” for its first appearance in the Introduction section of our revised manuscript.

2. Please make it explicit to the readership that the policy limiting the sale of cartridge-based e-cigarette products to only tobacco and menthol flavored e-cigarettes is the "FDA's flavor enforcement policy". 

Response: We have added the name of “the flavor enforcement policy” for the policy that limited the sale of cartridge-based e-cigarette tobacco and menthol flavors.

3. It is noted that "As menthol-flavored e-cigarettes gain popularity, menthol-flavored cigarettes are also becoming increasingly popular among youth and young adults, particularly in the African American community." More context needs to be provided for this statement as it does not fit with the rest of the paragraph and/or more information is needed to explain that this issue is a contributor to tobacco-related disparities (if that is the intent). Also, menthol-flavored cigarettes have been popular before menthol flavored e-cigarettes... so more needs to be clarified in these sentences. 

Response: Thank you for your feedback. We have revised the sentences to clarify the reasons for the emergence of synthetic cooling agents on the market. Additionally, we have omitted the discussion of tobacco-related disparities as it is not the primary focus of our study. The following are our revised sentences.

“Following the announcement and implementation of the FDA’s new policy, there was a significant increase in the sales of menthol-flavored e-cigarettes [11]. To curb the increasing popularity of menthol-flavored e-cigarettes, the New York State Department of Health banned all flavored vaping products, including menthol-flavored e-cigarettes, starting from May 18, 2020 [12]. Menthol-flavored cigarettes have been popular, with 40% of smokers using menthol-flavored cigarettes in 2020 [13]. In April 2022, the FDA announced the proposed standard to ban menthol–the last allowable flavor–in cigarettes in hopes of reducing tobacco-related diseases and deaths [14].

In response to the proposed menthol cigarette ban and potential national menthol-flavored e-cigarette ban, the market has seen a significant increase in 'ice' flavors that attempt to circumvent these regulations by incorporating synthetic cooling agents like WS-3 and WS-23 into e-cigarette refill liquids to mimic the cooling sensation like menthol or mint [15].”

4. Minor point: On page 5, line 98-99, missing "has" in "Overall, Twitter (has) served as an ideal..."

Response: We have inserted the missing “has” in “Overall, Twitter (has) served as an ideal…” in our revised manuscript.

METHODS:

1. The detail provided seems to be adequate.

Response: Thanks for the comments.

RESULTS:

1. Figures seem to blurry - authors may need to save the files in a different format and re-upload 

Response: We have saved the files in different format and re-uploaded them.

2. Rather than showing the percentage and the calculation for percentages, it might help to simplify the language and indicate that: "Among tweets with a positive attitude (n=2771), the most prevalent topic is enjoyment of ice vapes (64.49%), followed by liking lush ice vapes (18.0%) - for example. Also, the reporting of decimal places does not seem to be consistent.

Response: Thanks for the suggestions. We have removed calculations and simplified the language as suggested. We also made the decimal places consistent in our revised manuscript. 

“Among tweets with positive attitude (n = 2771), the most prevalent topic is the enjoyment of ice vapes (64.49%), followed by liking lush ice vapes (18.00%), liking the minty or icy feeling (7.32%), certain flavor being refreshing (3.10%), and enjoying the smell of vapes (0.79%).”

DISCUSSION:

1. It might be helpful if the authors can comment on whether any of the posts were #ads, as it would give more nuance regarding whether messaging is coming from vapers directly, or via the tobacco industry (for example).

Response: Thanks for the insightful comment. While we were trying to focus on messages from vapers only by removing all possible commercial tweets from the tobacco industry, by chance some tweets (advertisements) were not removed. We have added this as another limitation. “In addition, although we filtered out commercial content using related keywords in the preprocessing step, a small proportion of advertisement-related tweets, primarily sponsored YouTube review tweets and seller promotions that didn’t contain exclusion keywords, were included in the dataset. However, their presence was negligible and did not significantly impact the results of our analysis.”

2. It doesn't look like the discussion highlights a lot of other literature - is this a function of the approach being novel? If not, more citations are needed 

Response: Since the concept “synthetic cooling agents” is relatively new, there is not much literature on this. We have included as many relevant citations as we could.

3. Limitations as written is appropriate.

Response: Thanks for the comments.

Reviewer #2: The authors use machine learning to classify tweets with content about ice-type vape flavors. Classifications included valence, and whether the tweet was from a vaper or not.

Perhaps the most interesting finding was that the frequency of tweets that satisfied their search criteria more than doubled over the 20ish months of data. However, interpretation of this would be made easier if the reader was given comparison trajectories. Was the total tweets per month constant? What about the number of tweets about vaping in general – is this just reflecting overall change in interest, or, as plausibly suggested, a specific growth in interest in the WS-3/-23 agents.

Response: Thanks for the comments. We have included an example of a vaping-related conversation in response to an anti-vaping commercial. This conversation corresponds with increased discussions of synthetic cooling agents in May 2022, as shown in Figure 1. 

We have also included a supplementary figure (S2 Figure) showing the trend in the proportion of tweets related to synthetic cooling agents among all e-cigarette-related tweets, which showed the overall increase in the proportion of synthetic cooling agent-related tweets.

In addition to total interest as conveyed in tweet frequency, there are things I can imagine getting out of tweet content that would not be easily captured by questionnaire data. But the authors primarily focus on whether people do or do not like ice flavors. They conclude that they mostly do. But this seems a very indirect way of assessing how people feel about these products – it requires assumptions about what the relationship is to tweet content and general opinions. For example, I suppose people tweet more about things that they have strong opinions about. I have no idea whether they are more likely to tweet about things they like a lot or things they dislike. But vapes they like a lot are going to be more common in their day, so that might lead to over-representation of positive content. Or maybe people like to tweet views they think are less common. So my question is, what is it that these data are conveying that is not better conveyed by survey data and market data? There might be convincing answers to this that could be included.

Response: Thanks for these great comments! Knowing who is more likely to tweet positive or negative opinions toward synthetic cooling agents is very challenging. Therefore, there might be more tweets with positive or negative opinions, which could introduce certain biases. We have added this as one of the potential limitations in our study, “Moreover, people being more inclined to share their strong opinions on social media can lead to potential overrepresentation of users that have extremely negative or extremely positive sentiments.” 

As suggested, we discussed what social media data could provide compared to survey or market data. “As evidence, social media posts like these offer a rich source of qualitative data that traditional surveys may not capture. The immediate and personal nature of these posts provides valuable context. While there are many predictable tweets about frequently used products like popular vape flavors, Twitter also captures anecdotal views, which can provide a broader perspective. Unlike surveys, which typically present structured questions, social media data reveals spontaneous, unfiltered opinions and real-time discussions [39].”

The authors write, “Synthetic cooling agents in e-cigarettes imitate menthol's cooling sensation but induce a unique 'cold' feeling during inhalation, unlike the 'hot' sensation from capsaicin or menthol's 'cool' feeling.” I read the prior text to imply the WS-3 / -23 were intended to be similar to menthol to get around a potential ban. In what way is it “unlike” menthol’s cool feeling?

Response: We have added more description about the sensation of synthetic cooling agents, “Synthetic cooling agents in e-cigarettes imitate menthol's cooling sensation but induce a unique 'cold' feeling, akin to the feeling of inhaling frigid air, during inhalation, unlike the 'hot' burning or spicy sensation from capsaicin or menthol's 'cool', mild and refreshing mint feeling [17].”

---

## [Decision Letter · Decision Letter 1]

13 Feb 2024

Public perceptions of synthetic cooling agents in electronic cigarettes on Twitter

PONE-D-23-30383R1

Dear Dr. Li,

We’re pleased to inform you that your manuscript has been judged scientifically suitable for publication and will be formally accepted for publication once it meets all outstanding technical requirements.

Kind regards,

Gea Oliveri Conti, Ph.D. MBs

Academic Editor

PLOS ONE

Additional Editor Comments (optional):

Reviewers' comments:

Reviewer's Responses to Questions

**Comments to the Author**

1. If the authors have adequately addressed your comments raised in a previous round of review and you feel that this manuscript is now acceptable for publication, you may indicate that here to bypass the “Comments to the Author” section, enter your conflict of interest statement in the “Confidential to Editor” section, and submit your "Accept" recommendation.

Reviewer #1: All comments have been addressed

Reviewer #2: All comments have been addressed

2. Is the manuscript technically sound, and do the data support the conclusions?

Reviewer #1: Yes

Reviewer #2: Yes

3. Has the statistical analysis been performed appropriately and rigorously? 

Reviewer #1: Yes

Reviewer #2: Yes

4. Have the authors made all data underlying the findings in their manuscript fully available?

Reviewer #1: Yes

Reviewer #2: Yes

5. Is the manuscript presented in an intelligible fashion and written in standard English?

Reviewer #1: Yes

Reviewer #2: Yes

6. Review Comments to the Author

Reviewer #1: Thank you for carefully considering my suggestions for improvement. All of my comments have been adequately addressed.

Reviewer #2: I think the authors addressed the issues that could be addressed.

7. PLOS authors have the option to publish the peer review history of their article (what does this mean?). If published, this will include your full peer review and any attached files.

Reviewer #1: No

Reviewer #2: No

---

## [Editor Report · Acceptance letter]

1 Mar 2024

PONE-D-23-30383R1 

PLOS ONE

Dear Dr. Li, 

I'm pleased to inform you that your manuscript has been deemed suitable for publication in PLOS ONE. Congratulations! Your manuscript is now being handed over to our production team.

Kind regards, 

on behalf of

Dr. Gea Oliveri Conti 

Academic Editor

PLOS ONE